# Knowledge Transfer Based on Particle Filters for Multi-Objective Optimization

**Xilu Wang**  **and Yaochu Jin** * 

Faculty of Technology, Bielefeld University, 33619 Bielefeld, Germany
* Correspondence: yaochu.jin@uni-bielefeld.de

**Abstract:** Particle filters, also known as sequential Monte Carlo (SMC) methods, constitute a class of importance sampling and resampling techniques designed to use simulations to perform on-line filtering. Recently, particle filters have been extended for optimization by utilizing the ability to track a sequence of distributions. In this work, we incorporate transfer learning capabilities into the optimizer by using particle filters. To achieve this, we propose a novel particle-filter-based multi-objective optimization algorithm (PF-MOA) by transferring knowledge acquired from the search experience. The key insight adopted here is that, if we can construct a sequence of target distributions that can balance the multiple objectives and make the degree of the balance controllable, we can approximate the Pareto optimal solutions by simulating each target distribution via particle filters. As the importance weight updating step takes the previous target distribution as the proposal distribution and takes the current target distribution as the target distribution, the knowledge acquired from the previous run can be utilized in the current run by carefully designing the set of target distributions. The experimental results on the DTLZ and WFG test suites show that the proposed PF-MOA achieves competitive performance compared with state-of-the-art multi-objective evolutionary algorithms on most test instances.

**Keywords:** particle filter; multi-objective optimization; transfer learning

## 1. Introduction

Many real-world applications in economics, mechanics and engineering can be formulated as multi-objective optimization problems (MOPs) that simultaneously optimize two or more objective functions [1]. The basic statement of an MOP for a minimization task can be formulated as

$$min \quad \mathbf{F}(\mathbf{x}) = \{f_1(\mathbf{x}), f_2(\mathbf{x}), \cdots, f_m(\mathbf{x})\}$$
$$\mathbf{x} \subseteq \Omega \tag{1}$$

where $\Omega \subseteq \mathbb{R}^D$ is the decision space of decision variables, $\mathbf{x} = (x_1, x_2, \cdots x_D)$ is a decision vector with $D$ denoting the number of decision variables, $\mathbf{F}(\mathbf{x})$ consists of $m$ objective functions, and $m$ is the number of objectives.

Usually, different objectives are conflicting with each other, which means that a decision vector that decreases the values of $f_m$ may increases that of $f_n$. As a result, it is impossible to find only one solution that can optimize all the objectives simultaneously; however, a set of optimal solutions that trade off between different objectives are known as Pareto optimal solutions. The whole set of Pareto optimal solutions in the decision space is called the Pareto set (PS), and the projection of PS in the objective space is called the Pareto front. Various types of algorithms have been proposed for solving MOPs.

For example, the scalarization technique is one of the most popular methods and is used to convert an MOP into a single optimization problem. Scalarization can be achieved by the global criterion method [2], the weighted min-max method [3,4], the $\epsilon$-constraint method [5] and reference point methods [6].

Another popular approach is based on evolutionary algorithms (EAs), which have been applied successfully to many real-world complex optimization problems [7,8]. Over the past decades, a large number of multi-objective evolutionary algorithms (MOEAs) have been proposed, such as nondominated sorting genetic algorithm II (NSGA-II) [9], multi-objective evolutionary algorithm based on decomposition (MOEA/D) [10], reference vector guided evolutionary algorithm (RVEA) [11] and strength Pareto evolutionary algorithm 2 (SPEA2) [12]. More recently, many variants have been proposed to further enhance the optimization performance of MOEAs and extend them to many-objective optimization problems, such as NSGA-III [13], $\theta$-DEA [14] and MOEA/DD [15].

Particle filter (PF), also known as sequential Monte Carlo (SMC), is a class of importance sampling and resampling techniques designed to simulate from a sequence of probability distributions, and this has gained popularity over the last decade to solve sequential Bayesian inference problems. With the notable exception of linear-Gaussian signal-observation models, the PF theory has become the dominated approach to solving the state filtering problem in dynamic systems. Applications of particle filter theory have expanded to diverse fields, such as object tracking [16], navigation and guidance [17] and fault diagnosis [18].

Recently, particle filters have been extended for optimization [19,20] by utilizing the ability to track a sequence of distributions. In order to deal with a global optimization problem, generally, a sequence of artificial dynamic distribution is designed to employ the particle filter algorithm [21,22]. The crucial element in particle filter optimization (PFO) is how to design the system dynamic function by formulating the optimization problem as a filtering problem, which forces the set of particles to move toward the promising area containing optima.

Although PFO has shown promising performance in certain applications, current PFO methods only work for single-objective optimization problems [23]. As many real-world problems involve multiple objectives to be optimized simultaneously, it is interesting to extend PFO to MOPs. To fill this gap, we make an effort to extend the scope of the application of PFO to multi-objective cases. To achieve this, we propose a novel particle-filter-based multi-objective optimization algorithm (PF-MOA) by transferring knowledge acquired from the search experience.

The key insight adopted here is that, if we can construct a sequence of target distributions that can balance the multiple objectives and make the degree of the balance controllable, we can approximate the Pareto optimal solutions by simulating each target distribution via particle filters. Inspired by the ability of SMC samplers to sample sequentially from a sequence of probability distributions [24], we design a particle filter to perform the optimization. The method of importance updating in particle filters makes it possible to leverage the knowledge readily available for the previous subproblem to optimize the current subproblem, guiding the new particles to concentrate on the more promising area found thus far. As a result, PF-MOA offers an efficient solution to optimize MOPs by tracking the Pareto optimal solutions on the Pareto front via a particle filter.

The rest of this paper is organized as follows. Section 2 presents a brief introduction to particle filters and the application to single-objective optimization. In Section 3, a particle-filter-based multi-objective optimization method is proposed. Numerical simulations are conducted in Section 4, where the results are presented and discussed. Finally, our conclusions are drawn in Section 5.

## 2. Background

### 2.1. Particle Filter

Consider the discrete-time nonlinear state-space models relating a hidden state $x_k$ to the observations $y_k$:

$$x_k = g(x_{k-1}, u_k), k = 1, 2, \ldots,$$
$$y_k = h(x_k, v_k), k = 0, 1, \ldots, \tag{2}$$

where $k$ is the sample number; $x_k \in R^{n_x}$ is the state; $y_k \in R^{n_y}$ are the observations; $u_k \in R^{n_x}$ and $v_k \in R^{n_y}$ are the system and observation noise, respectively; and $n_x$ and $n_y$ are the dimensions of $x_k$ and $y_k$, respectively. We assume $u_k$ and $v_k$ are independent and identically distributed (i.i.d.) sequences, independent of each other and also independent of the initial state $x_0$, which has the probability density function (p.d.f.) $p_0$. Let $p(x_k \mid x_{k-1})$ denote the transition density, and $p(y_k \mid x_{k-1})$ denote the likelihood function.

The goal of filtering is to estimate the conditional density,

$$b_k(x_k) \triangleq p(x_k \mid y_{0:k}), \quad k = 0, 1, \ldots \tag{3}$$

where $y_{0:k} = \{y_0, \ldots, y_k\}$, for all the observations from time 0 to $k$. The conditional density $b_k(x_k)$ can be derived recursively via the Chapman–Kolmogorov equation and Bayes rule as follows:

$$
\begin{aligned}
b_k(x_k) &= \frac{p(y_k \mid x_k) p(x_k \mid y_{0:k-1})}{p(y_k \mid y_{0:k-1})} \\
&= \frac{p(y_k \mid x_k) \int p(x_k \mid x_{k-1}) b_{k-1}(x_{k-1}) dx_{k-1}}{\int p(y_k \mid x_k) p(x_k \mid y_{0:k-1}) dx_k}
\end{aligned}
\tag{4}
$$

Since $b_k(x_k)$ is unknown, we generate the particles by sampling from another known density $q(x_k \mid y_{0:k})$ and adjust the weights of the samples to obtain an estimate of $b_k(x_k)$. This approach is known as importance sampling, and the density $q(x_k \mid y_{0:k})$ is referred to as the importance density. Hence, it is easy to see that, in order to approximate $p(x_k \mid y_{0:k})$, for samples $\{x_k^i, i = 1, \ldots, N\}$ drawn i.i.d. from $q(x_k \mid y_{0:k})$, their weights should be

$$w_k^i \propto \frac{p(x_k^i \mid y_{0:k})}{q(x_k^i \mid y_{0:k})} \tag{5}$$

where $\propto$ means proportional to, and the weights should be normalized.

To perform the estimation recursively, we used the Bayes rule to derive the following recursive equation for the conditional density:

$$
\begin{aligned}
b_k(x_k) &\triangleq p(x_k \mid y_{0:k}) \\
&= \frac{p(x_k, y_k \mid y_{0:k-1})}{p(y_k \mid y_{0:k-1})} \\
&\propto p(y_k \mid x_k) \int p(x_k \mid x_{k-1}) p(x_{k-1} \mid y_{0:k-1}) dx_{k-1} \\
&\propto \int p(y_k \mid x_k) p(x_k \mid x_{k-1}) b_{k-1}(x_{k-1}) dx_{k-1}
\end{aligned}
\tag{6}
$$

where $p(y_k \mid y_{0:k-1}, x_k) = p(y_k \mid x_k)$ and $p(x_k \mid y_{0:k-1}, x_{k-1}) = p(x_k \mid x_{k-1})$ both follow from the Markovian property of model Equation (12), the denominator $p(y_k \mid y_{0:k-1})$ does not explicitly depend on $x_k$ and $k$, and $\propto$ means that $p(x_k \mid y_{0:k})$ is the normalized version of the right-hand side. The state transition density $p(x_k \mid x_{k-1})$ is induced from the state equation in Equation (12) and the distribution of the system noise $u_{k-1}$, and the likelihood $p(y_k \mid x_k)$ is induced from the observation equation in Equation (12) and the distribution of the observation noise $v_k$. Substituting Equation (6) into Equation (5), we find

$$w_k^i \propto \frac{p(y_k \mid x_k^i) p(x_k^i \mid x_{k-1}^i)}{q(x_k^i \mid y_{0:k})} p(x_{k-1}^i \mid y_{0:k-1}), \tag{7}$$

If the importance density $q(x_k \mid y_{0:k})$ is chosen to be factored as

$$q(x_k \mid y_{0:k}) = q(x_k \mid x_{k-1}, y_k) q(x_{k-1} \mid y_{0:k-1}) \tag{8}$$

Moreover, to avoid sample degeneracy, new samples are resampled i.i.d. from the approximate conditional density $\hat{p}(x_k \mid y_{0:k})$ at each step; hence, the weights are reset to $w_{k-1}^i = 1/N$, and

$$w_k^i \propto \frac{p\left(y_k \mid x_k^i\right) p\left(x_k^i \mid x_{k-1}^i\right)}{q\left(x_k^i \mid x_{k-1}^i, y_k\right)}, i = 1, \ldots, N \tag{9}$$

In the plain particle filter, the importance density $q\left(x_k \mid x_{k-1}^i, y_k\right)$ is chosen to be the state transition density $p\left(x_k \mid x_{k-1}^i\right)$, which is independent of the current observation $y_k$, yielding

$$w_k^i \propto p\left(y_k \mid x_k^i\right), i = 1, \ldots, N \tag{10}$$

The plain particle filter recursively propagates the support points and updates the associated weights. The algorithm is as follows in Algorithm 1:

---

**Algorithm 1** General particle filter.

---

1: Initialization: Sample $\{x_0^i\}_{i=1}^N$ i.i.d. from an initial p.d.f. $p_0$. Set $k = 1$.
2: Importance Sampling/Propagation: Sample $x_k^i$ from $p(x_k \mid x_{k-1}^i), i = 1, \ldots, N$.
3: Bayes Updating: Receive new observation $y_k$. The conditional density is approximated by $\hat{p}(x_k \mid y_{0:k}) = \sum_{i=1}^N w_k^i \delta(x - x_k^i)$, where $w_k^i$ is computed according to Equation (10).
4: Resampling: Sample $\{x_k^i\}_{i=1}^N$ i.i.d. from $\hat{p}(x_k \mid y_{0:k})$.
5: $k \leftarrow k + 1$ and go to step 2.

---

### 2.2. Particle Filter Optimization for Global Optimization

We consider the global optimization problem:

$$x^* \in \underset{x \in \mathcal{X}}{\arg\max} H(x) \tag{11}$$

where $x$ is a vector of n decision variables, $\mathcal{X}$ is the search space, and the objective function $H$ is a bounded deterministic function. We denote the optimal function value as $H^*$, i.e., there exists an $x^*$ such that $H(x) \leq H^* \triangleq H(x^*), \forall x \in \mathcal{X}$.

Many of the simulation-based global optimization methods, such as the estimation of distribution algorithms (EDAs) [25,26], covariance matrix adaptation evolution strategy [27], cross-entropy (CE) method [28], model reference adaptive search (MRAS) method [29] and particle filter optimization (PFO), fall into the category of model-based methods. They share the similarities of iteratively repeating the following two steps: let $g_k$ be a probability distribution on $x$ at the $k$-th iteration of an algorithm:

- Randomly generate a set/population of candidate solutions $X^{(k)}$ from an intermediate distribution $g_k$ over the solution space.
- Update the intermediate distribution $g_k$ using the candidate solutions to obtain a new distribution $g_{k+1}$; increase $k$ by 1 and reiterate from step 1.

The underlying idea is to construct a sequence of iterates (probability distributions) $g_k$ with the hope that $g_k \to g^*$ as $k \to \infty$, where $g^*$ is a limiting distribution that assigns most of its probability mass to the set of optimal solutions. Thus, it is the probability distribution (as opposed to candidate solutions as in instance-based algorithms) that is propagated from one iteration to the next [30].

The main idea of PFO is to formulate the optimization problem as a filtering problem, then particle filter construction appears as a natural candidate for the reformulation of the global optimization problem as a filtering problem. More specifically, the optimiza-

tion problem Equation (11) can be formulated as a filtering problem by constructing an appropriate state-space model. Let the state-space model be

$$
\begin{aligned}
x_k &= x_{k-1}, \quad k = 1, 2, \ldots, \\
y_k &= H(x_k) - v_k, \quad k = 0, 1, \ldots,
\end{aligned}
\tag{12}
$$

where the optimal solution is a static state to be estimated, $x_k$ is the unobserved state, $y_k$ is the observation, $v_k$ is the observation noise that is an i.i.d. sequence, and the conditional density of the state approaches a delta function concentrated on the optimal solution as the system evolves.

We assume that $v_k$ has a p.d.f. $\varphi(\cdot)$, and then the transition density is

$$
p(x_k \mid x_{k-1}) = \delta(x_k - x_{k-1})
\tag{13}
$$

where $\delta$ denotes the Dirac delta function. The likelihood function is

$$
\begin{aligned}
p(y_k \mid x_k) &= \varphi(H(x_k) - y_k) \\
&= \varphi(H(x_{k-1}) - y_k)
\end{aligned}
\tag{14}
$$

Substituting Equations (13) and (14) into the recursive equation of conditional density Equation (6), we obtain

$$
b_k(x_k) = \frac{\varphi(H(x_k) - y_k)b_{k-1}(x_k)}{\int \varphi(H(x_k) - y_k)b_{k-1}(x_k)dx_k}
\tag{15}
$$

The intuition of model Equation (12) is that the optimal solution $x^*$ is an unobserved static state, while we can only observe the optimal function values $y^* = H(x^*)$ with some noise. Equation (15) implies that, at each iteration, the conditional density (i.e., $b_{k-1}$) is tuned by the performance of solutions to yield a new conditional density (i.e., $b_k$) for drawing candidate solutions at the next iteration.

It should be expected that, if $y_k$ increases with $k$, the conditional density $b_k$ will come closer to the density of $x_k$, i.e., a Dirac delta function concentrated on $x^*$. From the viewpoint of filtering, $b_k$ is the posterior density of $x_k$ that approaches the density of $x_k$. From the optimization viewpoint, $b_k$ is a density defined on the solution space that becomes increasingly concentrated on the optimal solution as $k$ increases. The framework of general particle filter optimization is given in Algorithm 2.

---

**Algorithm 2** General particle filter optimization framework.

---

1: Initialization: Sample $\{x_0^i\}_{i=1}^N$ i.i.d. from an initial p.d.f. $p_0$. Set $k = 1$.
2: Importance Sampling/Propagation: Sample $x_k^i$ from $p(x_k \mid x_{k-1}^i), i = 1, \ldots, N$.
3: Bayes Updating: Take $y_k$ to be the sample function value of $H(x_k^i)$ according to a certain rule. Compute the weight $w_k^i$ for sample $x_k^i$ according to $w_k^i \propto \varphi(H(x_k^i) - y_k), i = 1, 2, \ldots, N_k$ and normalize the weights such that they sum up to 1.
4: Resampling: Sample $\{x_k^i\}_{i=1}^N$ i.i.d. from $\hat{p}(x_k \mid y_{0:k})$.
5: $k \leftarrow k + 1$ and go to step 2.

---

Generally, the PFO algorithms can be differentiated from each other by the definitions of the target p.d.f. and of the proposal p.d.f. A specific definition of the target and proposal p.d.f. determines how the objective function is implanted in the sampling process and how the random samples (i.e., candidate solutions) are generated, respectively. For example, while a uniform distribution is adopted as the likelihood function $p(y_k \mid x_{k-1})$ in [22,31], the Boltzmann distribution is another choice in defining the target distribution for PFO methods [21].

## 3. Proposed Algorithm

### 3.1. Algorithm Framework

As mentioned above, particle filter optimization methods have been applied to single-objective optimization problems by reformulating an optimization problem into a filtering problem. In this work, we make an effort to extend the scope of application of PFO to multi-objective cases. It is well-known that a Pareto optimal solution to a MOP, under mild conditions, could be an optimal solution of a scalar optimization problem in which the objective is an aggregation of all the objectives [10]. That is to say, MOPs can be formulated as a task of searching a set of Pareto optimal solutions, each of which corresponds to a scalar optimization subproblem with a certain degree of tradeoff among the objectives in an MOP.

With the insight into the decomposition strategy in the context of MOPs, it makes sense to construct a series of target distributions corresponding to a number of scalar objective optimization subproblems, and then the particle filter is adopted to simulate these distributions so that the Pareto optimal solutions can be obtained based on the samples yielded from simulations. There are two main issues: (1) how to design a series of proxy target pdfs for MOPs and (2) how to effectively simulate these p.d.fs via SMC. In the following, we seek answers to these problems and propose a particle filter optimization method for solving multiobjective optimization problems, which will be elaborated in the following. The framework of the proposed PF-MOA is outlined in Algorithm 3.

---

**Algorithm 3** Particle filter multiobjective optimization.

**Input:** $N$: the number of particles; $K$: the number of subproblems; set the maximum number of fitness evaluation $FE^{max} = N * K$;

**Output:** particles in the archive $D$;

1: Initialization: Generate a uniform spread of $K$ weight vectors $\lambda$ for the Tchebycheff approach; optimize the first subproblem and obtain $N$ particles $\{x_0^i\}_{i=1}^N$ to be evaluated on objective functions; save the particles in $D$; set $z^* = (z_1^*, ..., z_K^*)$ with $z_i^* = min f_i(\mathbf{x})$; set $k = 1$ and $FE = N$;

2: **while** $FE \leqslant FE^{max}$ **do**

3:     **for** $k = 1, \ldots, K$ **do**

4:         //Computing the $k$-th subproblem//

5:         Calculate the subproblem with the $k$-th weight vector according to Equation (16): $g^{tch}(\mathbf{x} \mid \lambda^k, z^*) = \max_{1 \leq i \leq m} \{\lambda_i^k (f_i(\mathbf{x}) - z_i^*)\}$.

6:         Update the reference point $z^*$.

7:         //Computing the target pdf associated with the $k$-th subproblem//

8:         Calculate the corresponding $k$-th target pdf according to Equation (17): $\tilde{\pi}_k(\mathbf{x}) \triangleq \frac{\pi_k(\mathbf{x})}{C_k}$ and $\pi_k(\mathbf{x}) = \exp\{-g^{tch}\}$.

9:         **for** $i = 1, \ldots, N$ **do**

10:             //Importance Updating//

11:             Compute the weight $\hat{\omega}_k^i$ for each sample $\mathbf{x}_k^i$ according to Equation (18), $\hat{\omega}_k^i = \begin{cases} \tilde{\pi}_k(\mathbf{x}_k^i), & \text{if} \quad k = 1 \\ \tilde{\pi}_k(\mathbf{x}_k^i)/\tilde{\pi}_{k-1}(\mathbf{x}_k^i), & \text{otherwise}. \end{cases}$

12:         **end for**

13:         Normalize the weights such that they sum up to 1;

14:         Resampling: Generate $N$ i.i.d. samples by setting $\bar{\mathbf{x}}^i = \mathbf{x}_k^j$ with probability $\hat{\omega}_k^j, j = 1, \ldots, N$. Then, set $\mathbf{x}_k^i = \bar{\mathbf{x}}^i, \hat{\omega}_k^i = 1/N$, for $\forall i$.

15:         Calculate the mean of particles $\bar{\mathbf{x}}_k$ and the best particle $\mathbf{x}_k^\star$ obtained thus far.

16:         Particle move: Generate new particles $\mathbf{x}'$ using genetic operators on $\bar{\mathbf{x}}_k$ and $\mathbf{x}_k^\star$, shown in Algorithm 4.

17:         Update the reference point $z^*$.

18:         Save the particles $\{\mathbf{x}_k^i\}_{i=1}^N$ to $D$.

19:     **end for**

20:     Update $k = k + 1$, $FE = FE + N$;

21: **end while**

22: Return the particles in $D$;

---

### 3.2. The Design of Target Distribution

Based on the theoretical foundation of sequential Monte Carlo samplers [24], SMC allows us to perform global optimization and sequential Bayesian estimation by sequentially sampling from a sequence of probability distributions that are defined on a common space. Specifically, similar to simulated annealing [32], we can move from a tractable distribution

to a distribution of interest through a sequence of artificial intermediate distributions. Consequently, the convergence results are available for SMC samplers [33]. As two or more conflicting objectives are involved in an MOP in Equation (1), the design of the target p.d.f. is different from that in single-objective optimization problems.

To approach to the Pareto optimal set, a set of proxy target pdfs are needed, each of which corresponding to a specific amount of balance among the objectives. To this end, we adopted a decomposition strategy to decompose an MOP into a number of scalar optimization subproblems, followed by designing a target p.d.f. for each single-objective subproblem. More specifically, let $\lambda^1, ..., \lambda^K$ be a set of even spread weight vectors, and let $\mathbf{z}^*$ be the reference point. An MOP with $m$ objectives, i.e., Equation (1), can be decomposed into $K$ scalar/single-objective optimization subproblems using the Tchebycheff (TCH) decomposition [10], and the objective function of the $j$th subproblem is

$$\min_{\mathbf{x} \in \Omega} g^{\text{tch}}\left(\mathbf{x} \mid \lambda^j, \mathbf{z}^*\right) = \max_{1 \le i \le m}\left\{\lambda_i^j(f_i(\mathbf{x}) - z_i^*)\right\} \tag{16}$$

where $m$ is the number of objectives, $\mathbf{z}^* = (z_1^*, ..., z_m^*)$ with $z_i^* = \min f_i(\mathbf{x} \mid \mathbf{x} \in \Omega)$ is the reference point, $\lambda^j = \left(\lambda_1^j, ..., \lambda_m^j\right)$ with $\sum_{i=1}^m \lambda_i = 1$ and $\lambda_i \ge 0$ is the weight vector, and $f_i$ and $\mathbf{x}$ are the objective function and decision vector, respectively.

In this way, for each Pareto optimal solution $\mathbf{x}^*$ of an MOP, there exists a weight vector $\lambda$ such that $\mathbf{x}^*$ is the optimal solution of a subproblem (Equation (16)), and each optimal solution of the subproblem is Pareto optimal to the MOP. As a result, to obtain a set of different Pareto optimal solutions of an MOP, one can solve a set of single-objective optimization problems with different weight vectors defined by Equation (16) or any other decomposition approaches. Note that $g^{\text{tch}}$ is continuous of $\lambda$, the optimal solution of $g^{\text{tch}}\left(\mathbf{x} \mid \lambda^i, \mathbf{z}^*\right)$ should be close to that of $g^{\text{tch}}\left(\mathbf{x} \mid \lambda^j, \mathbf{z}^*\right)$ if $\lambda^i$ and $\lambda^j$ are close to each other. Therefore, any information about these $g^{\text{tch}}$ with weight vectors close to $\lambda^i$ should be helpful for optimizing $g^{\text{tch}}\left(\mathbf{x} \mid \lambda^i, \mathbf{z}^*\right)$.

Obtaining a set of single-objective subproblems, a set of target p.d.fs $\tilde{\pi}_1(\mathbf{x}), \tilde{\pi}_2(\mathbf{x}), ...,$ $\tilde{\pi}_K(\mathbf{x})$ corresponding to the subproblems are constructed as follows,

$$\tilde{\pi}_k(\mathbf{x}) \triangleq \frac{\pi_k(\mathbf{x})}{C_k}, k = 1, 2, ..., K$$
$$\pi_k(\mathbf{x}) = \exp\left\{-g^{\text{tch}}\right\} \tag{17}$$

where $K$ is the number of target p.d.fs (in our case, $K$ equals to the number of the weight vector), $C_k$ is a normalizing constant which ensures $\tilde{\pi}_k(\mathbf{x})$ to be a qualified pdf whose integral equals 1. According to Equations (16) and (17), each p.d.f. corresponds to a specific degree of balance between each objective using the weight vectors.

### 3.3. The Sampling Procedure

Given the target p.d.fs, the particle filter appears as a natural candidate for the simulation of these target distributions. The first subproblem is optimized, and then the particle filter is used to track the sequence of target distributions that correspond to a set of scalar subproblems. This has three main steps: importance updating, resampling and particle move. The importance updating step takes the current distribution $\tilde{\pi}_k$ (corresponding to a subproblem) as the target distribution and takes the previous distribution $\tilde{\pi}_{k-1}$ (corresponding to the previous subproblem) as the proposal distribution.

Thus, given that the previous samples are updated in proportion to $\tilde{\pi}_k(\cdot)/\tilde{\pi}_{k-1}(\cdot)$, the new empirical distribution formed by samples is already distributed approximately according to $\tilde{\pi}_{k-1}$, and the weights of these weighted samples will closely follow $\tilde{\pi}_k$. The resampling step redistributes the samples such that they all have equal weights. The particle move step is performed on each particle to update their locations towards the

promising region so that we can follow the target distribution of each subproblem as closely as possible.

Note that, instead of updating particles according to a transition equation as in Equation (12), a Metropolis sampling method associated with genetic operators is adopted to sample new particles as the transition equation in the MOPs is unknown.

From the perspective of multi-objective optimization, the advantage of the proposed PF-MOA can be explained by tracking the Pareto optimal solutions on the Pareto front and making the search more efficient. The reason is that the importance weight of particles in the proposed PF-MOA is updated according to the difference between the current and the previous distributions (which correspond to two related subproblems). As we mentioned in Section 3.2, any information about these $g^{\text{tch}}$ with weight vectors close to $\boldsymbol{\lambda^i}$ should be helpful for optimizing $g^{\text{tch}}(\mathbf{x} \mid \boldsymbol{\lambda^i}, \mathbf{z}^*)$. The method of importance updating makes it possible to leverage the knowledge readily available for the previous subproblem to optimize the next subproblem, guiding the new particles to concentrate on the more promising area found thus far.

More specifically, while the normalization of the weights and the resampling of the particles are the typical operations in Algorithm 2, the calculation of the importance weight for the $i$-th particle according to the set of target distributions is as follows,

$$\hat{\omega}_k^i = \begin{cases} \tilde{\pi}_k(\mathbf{x}^i), & \text{if} \quad k = 1 \\ \tilde{\pi}_k(\mathbf{x}^i) / \tilde{\pi}_{k-1}(\mathbf{x}^i), & \text{otherwise.} \end{cases} \tag{18}$$

Through the resampling step, we eliminate/duplicate samples with low/high importance weights, respectively, avoiding the issue of particle degeneracy.

*3.4. Particle Move*

After the resampling step, a Metropolis sampling method based on genetic operators is proposed to promote the divergence of particles as summarized in Algorithm 4. As we demonstrate in Section 2.2, the state transition as function is assumed $x_k = x_{k-1}$ in the state space model when solving global optimization problems. If a particle filter is applied to this model directly, with no particle move, the resulting algorithm would be equivalent to importance sampling from the initial sampling distribution directly to the posterior in a single step. This would be problematic if the initial sampling distribution was located in a different region of parameter space entirely, particularly in the context of MOPs. Hence, a Metropolis sampling method for generating new particles is proposed to assist the particle filter to simulate these target pdfs by exploiting the promising region.

The resampling step together with the Metropolis sampling step prevents sample degeneracy or, in other words, maintains the sample diversity and, thus, the exploration of the solution space. To make use of the search information obtained by the particle filter, the mean of particles $\bar{x}_k$ and the best particle $\mathbf{x}_k^\star$ obtained thus far are identified and assumed to be close to the optimum. The new/displacement particles $\mathbf{x}'$ will hence be generated around the promising region using genetic operators, i.e., the typical mutation and crossover operators. Subsequently, the displacement will be either accepted or rejected according to a dynamically calculated probability, called the acceptance probability. In the proposed PF-MOA, the acceptance probability for the displacement of the $i$-th particle $(\mathbf{x}^i)$ is calculated by

$$\rho = \min\left\{ \tilde{\pi}_k(\mathbf{x}') / \tilde{\pi}_k(\mathbf{x}^i), 1 \right\}. \tag{19}$$

---

**Algorithm 4** A Metropolis sampling method based on genetic operators.

---

**Input:** The current particles $\{\mathbf{x}_k^i\}_{i=1}^N$ and the current target pdf $\tilde{\pi}_k$, the mean of particles $\bar{x}_k$ and the best particle $\mathbf{x}_k^\star$ obtained thus far.

**Output:** the new particles;

1: **for** $i = 1 : N$ **do**
2:      Perform the genetic operator on $\bar{x}_k$ and $\mathbf{x}_k^\star$ and generate a new particle $\mathbf{x}'$.
3:      Calculate acceptance probability via Equation (19) and replace $\mathbf{x}_k^i$ by $\mathbf{x}'$ with

$$\mathbf{x}_k^i = \begin{cases} \mathbf{x}', & \text{with probability } \rho \\ \mathbf{x}_k^i, & \text{with probability } 1 - \rho \end{cases} \tag{20}$$

4:      Update $\mathbf{x}_k^\star = \mathbf{x}'$, if $\tilde{\pi}_k(\mathbf{x}') > \tilde{\pi}_k(\mathbf{x}_k^\star)$.
5: **end for**
6: Return updated particles;

---

## 4. Comparative Studies

In this section, numerical experiments are conducted on nine three-objective benchmark problems taken from the DTLZ test suite. To examine the efficiency of the proposed strategies, the proposed PF-MOA is compared with state-of-the-art multi-objective evolutionary algorithms, NSGA-II [9], RVEA [11], MOEA/D [10], NSGA-III [13], MOEA/DD [15] and $\theta$-DEA [14]. Our code is available at https://github.com/xw00616/PF-MOA (accessed on 1 November 2022).

In the following section, we begin with briefly introducing the test problems and performance metrics adopted in our paper. Afterwards, the details of the experimental settings concerning the four compared algorithms are described. Lastly, the experimental results together with the Wilcoxon rank sum test are presented and discussed.

### 4.1. Test Problems

In our experiments, the proposed algorithm is compared with three state-of-the-art multi-objective optimization algorithms on DTLZ [34] and WFG [35] test suites with three objectives. The number of decision variables for the DTLZ test instances is set to $D = M + K - 1$, where $K = 5$ is adopted for DTLZ1, $K = 10$ is used for DTLZ2 to DTLZ6, and $K = 20$ is employed in DTLZ7. The number of decision variables for the WFG test instances is set to 12. $M$ represents the number of objectives; here, we set $M = 3$.

### 4.2. Performance Metrics

The inverted generational distance (IGD) [36] metric and hypervolume (HV) [37] metric are adopted to assess the performance of the algorithms. IGD and HV provide a combined information of the convergence and diversity of the obtained set of solutions. The PlatEMO toolbox [38] is used to calculate values of the performance metric in our experiments. Let $P^*$ be a set of uniformly distributed solutions sampled from objective space along the theoretical Pareto front. Let $P$ be an obtained approximation to the Pareto front. Let $P^*$ be a set of uniformly distributed solutions sampled from objective space along the theoretical Pareto front. IGD measures the inverted generational distance from $P^*$ to $P$, defined as

$$IGD(P^*, P) = \frac{\sum_{v \in P^*} d(v, P)}{|P^*|} \tag{21}$$

where $d(v, P)$ is the minimum Euclidean distance between $v$ and all points in $P$. The smaller IGD value, the better the achieved solution set is.

HV calculates the volume of the objective space dominated by an approximation set $P$ and dominates $P^*$ sampled from the PF.

$$HV = \text{volume}\left(\cup_{i=1}^j \vartheta_i\right) \tag{22}$$

where $\vartheta_i$ represents the hypervolume contribution of the $i$-th solutions relative to the reference points. All HV values presented in this paper are normalized to $[0, 1]$. Algorithms achieving a larger HV value are better.

### 4.3. Experimental Settings

We ran each algorithm on each benchmark problem 20 independent times, and the Wilcoxon rank sum test was calculated to compare the mean of 20 running results obtained by PF-MOA and by the compared algorithms at a significance level of 0.05. Symbols "(–)", "(+)" and "($\approx$))" indicate that the proposed algorithm shows significantly better, worse and similar performance than the compared algorithm, respectively.

The PF-MOA was implemented in MATLAB R2019a on an Intel Core i7 with 2.21 GHz CPU, and the compared algorithms were implemented in PlatEMO toolbox [38]. The general parameter settings in the experiments are given as follows: (1) The maximum number of function evaluations $FE_{max} = 10,000$. (2) For PF-MOA: the population size was set to 100 and the maximum number of generations was set to 100. (3) For the three multiobjective evolutionary algorithms: the population size was set to 100 and the maximum number of generations was set to 100. The specific parameter settings for each compared algorithm were the same as recommended in their original papers.

### 4.4. Experimental Results

The statistical results in terms of IGD and HV values obtained by the four algorithms are summarized in Table 1 and Table 2, respectively. For the DTLZ test problems, it is apparent that the proposed PF-MOA achieved the best approximate Pareto front on all test problems except for DTLZ6 and DTLZ7 (NSGA-II obtained the best IGD values). The reason behind this may be that DTLZ6 has a plenty of disconnected Pareto optimal regions in the decision space, and DTLZ7 has a discontinuous Pareto front. Hence, it is challenging to design proper target distributions in PF-MOA, which further degrades PF-MOA's performance.

According to the Wilcoxon rank sum test, the proposed algorithm significantly outperformed the compared algorithms on most of the test problems. For the WFG test instances, PF-MOA showed significantly better performance than the algorithms under comparison on six out of nine test instances, confirming the promising performance of the proposed PF-MOA. More specifically, taking WFG5 as an example, the objective multimodality was combined with landscape deception, and the proposed PF-MOA showed the worst performance compared with the other algorithms.

A possible explanation for this is that the deceptive objectives may impact the design of the target distributions, and the information form the previous subproblem does not provide sufficient information to help the algorithm generate good tradeoff solutions for the current subproblem. Moreover, similar observations can be made from Table 2.

To further illustrate the performance of the proposed algorithm, the obtained Pareto front for each algorithm is illustrated in Figure 1. We observed that the proposed method can find a set of well-converged and diverse Pareto optimal solutions, thereby, confirming the effectiveness of the particle filter in the PF-MOA.

**Table 1.** Statistical results of the IGD values obtained by NSGA-II, RVEA, MOEA/D, MOEA/DD, NSGA-III, $\theta$-DEA and PF-MOA with the same number of real function evaluations.

| Problem | D | MOEA/D | NSGA-II | RVEA | NSGA-III | MOEA/DD | $\theta$-DEA | PF-MOA |
|---|---|---|---|---|---|---|---|---|
| DTLZ1 | 7 | 2.41e-1 (3.56e-1) − | 4.00e-1 (4.17e-1) − | 4.67e-1 (2.83e-1) − | 1.86e-1 (1.73e-1) ≈ | 3.51e-1 (2.36e-1) − | 2.22e-1 (3.58e-1) − | 1.27e-1 (7.06e-2) |
| DTLZ2 | 12 | 5.48e-2 (2.21e-4) − | 6.96e-2 (2.44e-3) − | 5.59e-2 (6.45e-4) − | 5.51e-2 (2.45e-4) − | 5.54e-2 (1.95e-4) − | 5.48e-2 (4.76e-5) − | 4.11e-2 (1.01e-2) |
| DTLZ3 | 12 | 1.18e+1 (6.40e+0) − | 1.02e+1 (6.66e+0) − | 1.63e+1 (5.65e+0) − | 1.04e+1 (2.75e+0) − | 2.22e+1 (1.07e+1) − | 1.06e+1 (1.10e+0) − | 1.79e+0 (3.79e+0) |
| DTLZ4 | 12 | 4.89e-1 (3.50e-1) + | 1.15e-1 (1.44e-1) + | 5.59e-2 (5.89e-4) + | 5.51e-2 (1.20e-4) + | 5.55e-2 (7.39e-4) + | 5.49e-2 (7.99e-5) + | 6.48e-1 (2.32e-1) |
| DTLZ5 | 12 | 3.23e-2 (7.34e-4) − | 6.10e-3 (3.36e-4) − | 8.45e-2 (1.70e-2) − | 1.27e-2 (2.16e-3) − | 3.13e-2 (1.03e-3) − | 3.01e-2 (2.18e-3) − | 3.87e-3 (1.21e-3) |
| DTLZ6 | 12 | 2.09e-1 (3.93e-1) + | 3.58e-2 (1.61e-1) + | 1.46e-1 (1.42e-1) + | 1.89e-2 (2.42e-3) + | 1.00e-1 (1.39e-1) + | 3.81e-2 (1.71e-3) + | 7.63e+0 (3.62e-1) |
| DTLZ7 | 22 | 2.19e-1 (1.99e-1) + | 1.13e-1 (6.90e-2) + | 2.02e-1 (5.32e-2) + | 1.61e-1 (1.39e-1) + | 4.29e-1 (2.44e-1) + | 9.62e-2 (5.39e-3) + | 7.45e+0 (6.91e-1) |
| WFG1 | 12 | 6.84e-1 (1.00e-1) + | 5.72e-1 (8.19e-2) + | 7.62e-1 (8.89e-2) + | 1.03e+0 (3.60e-2) ≈ | 1.51e+0 (6.12e-2) − | 1.01e+0 (4.76e-2) ≈ | 1.82e+0 (1.41e-1) |
| WFG2 | 12 | 3.33e-1 (7.77e-2) − | 2.24e-1 (9.68e-3) − | 2.16e-1 (1.22e-2) − | 1.75e-1 (7.12e-3) − | 1.91e-1 (9.10e-3) − | 1.60e-1 (2.81e-3) − | 2.03e-2 (2.65e-2) |
| WFG3 | 12 | 3.55e-1 (1.24e-1) − | 1.29e-1 (2.21e-2) − | 2.65e-1 (2.41e-2) − | 1.65e-1 (1.40e-2) ≈ | 3.79e-1 (1.12e-1) − | 1.43e-1 (1.99e-2) + | 2.18e-1 (1.51e-2) |
| WFG4 | 12 | 2.91e-1 (1.25e-2) − | 2.82e-1 (1.04e-2) − | 2.69e-1 (7.74e-3) − | 2.32e-1 (1.67e-3) − | 2.48e-1 (3.02e-3) − | 2.29e-1 (1.10e-3) − | 1.60e-1 (2.68e-2) |
| WFG5 | 12 | 2.73e-1 (9.26e-3) + | 2.85e-1 (1.22e-2) + | 2.61e-1 (7.34e-3) + | 2.37e-1 (2.13e-3) + | 2.52e-1 (1.05e-3) + | 2.36e-1 (1.38e-3) + | 6.87e-1 (2.43e-2) |
| WFG6 | 12 | 3.44e-1 (2.25e-2) − | 3.29e-1 (1.92e-2) − | 3.37e-1 (2.06e-2) − | 2.78e-1 (2.12e-2) − | 3.04e-1 (2.23e-2) − | 2.67e-1 (1.35e-2) − | 1.71e-1 (3.38e-2) |
| WFG7 | 12 | 4.17e-1 (5.34e-2) ≈ | 2.83e-1 (1.04e-2) ≈ | 2.89e-1 (1.33e-2) ≈ | 2.32e-1 (1.13e-3) ≈ | 2.66e-1 (1.56e-2) ≈ | 2.29e-1 (8.96e-4) ≈ | 2.63e-1 (5.54e-2) |
| WFG8 | 12 | 3.78e-1 (2.51e-2) + | 3.75e-1 (1.02e-2) + | 3.74e-1 (1.39e-2) + | 3.20e-1 (5.91e-3) + | 3.37e-1 (7.67e-3) + | 3.16e-1 (6.32e-3) + | 5.48e-1 (3.59e-2) |
| WFG9 | 12 | 3.70e-1 (6.98e-2) − | 2.84e-1 (2.24e-2) − | 2.78e-1 (3.30e-2) − | 2.38e-1 (3.51e-3) − | 2.61e-1 (1.63e-2) − | 2.35e-1 (4.19e-3) − | 1.21e-1 (2.11e-2) |
| +/−/≈ | | 6/10/0 | 7/8/1 | 6/10/0 | 5/7/4 | 5/10/1 | 6/8/2 | |

**Table 2.** Statistical results of the HV values obtained by NSGA-II, RVEA, MOEA/D, MOEA/DD, NSGA-III, $\theta$-DEA and PF-MOA with the same number of real function evaluations.

| Problem | D | MOEA/D | NSGA-II | RVEA | NSGA-III | MOEA/DD | $\theta$-DEA | PF-MOA |
|---|---|---|---|---|---|---|---|---|
| DTLZ1 | 7 | 6.21e-1 (3.49e-1) − | 3.60e-1 (4.02e-1) − | 2.12e-1 (2.86e-1) − | 4.57e-1 (3.40e-1) − | 2.27e-1 (3.15e-1) − | 5.78e-1 (3.33e-1) − | 9.28e-1 (7.50e-2) |
| DTLZ2 | 12 | 5.55e-1 (6.99e-4) − | 5.29e-1 (7.43e-3) − | 5.51e-1 (2.97e-3) − | 5.55e-1 (6.66e-4) − | 5.54e-1 (8.13e-4) − | 5.56e-1 (5.04e-4) − | 6.91e-1 (1.01e-4) |
| DTLZ3 | 12 | 0.00e+0 (0.00e+0) − | 0.00e+0 (0.00e+0) − | 0.00e+0 (0.00e+0) − | 0.00e+0 (0.00e+0) − | 0.00e+0 (0.00e+0) − | 0.00e+0 (0.00e+0) − | 2.00e-2 (1.27e-2) |
| DTLZ4 | 12 | 2.91e-1 (1.12e-1) ≈ | 4.96e-1 (8.42e-2) + | 5.53e-1 (1.93e-3) + | 5.56e-1 (8.24e-4) + | 5.55e-1 (1.45e-3) + | 5.55e-1 (4.87e-4) + | 2.48e-1 (1.32e-1) |
| DTLZ5 | 12 | 1.82e-1 (4.00e-4) − | 1.98e-1 (3.78e-4) − | 1.48e-1 (4.11e-3) − | 1.93e-1 (1.91e-3) − | 1.82e-1 (2.29e-4) − | 1.83e-1 (6.22e-4) − | 4.87e-1 (2.51e-4) |
| DTLZ6 | 12 | 1.77e-1 (3.83e-3) + | 1.99e-1 (1.59e-4) + | 1.35e-1 (2.14e-2) + | 1.90e-1 (1.28e-3) + | 1.54e-1 (4.42e-2) + | 1.81e-1 (1.71e-3) + | 0.00e+0 (0.00e+0) |
| DTLZ7 | 22 | 2.32e-1 (9.16e-3) + | 2.47e-1 (3.41e-3) + | 2.06e-1 (2.69e-2) + | 2.39e-1 (1.33e-2) + | 2.11e-1 (1.70e-2) + | 2.53e-1 (1.69e-3) + | 0.00e+0 (0.00e+0) |
| WFG1 | 12 | 3.72e-1 (2.30e-2) − | 4.91e-1 (4.72e-2) ≈ | 4.58e-1 (2.91e-2) − | 4.60e-1 (1.22e-2) ≈ | 2.82e-1 (5.79e-3) − | 4.61e-1 (1.74e-2) ≈ | 4.78e-1 (1.41e-2) |
| WFG2 | 12 | 8.20e-1 (3.14e-2) − | 9.07e-1 (6.22e-3) − | 8.93e-1 (1.12e-2) − | 9.09e-1 (2.33e-3) − | 8.93e-1 (1.04e-2) − | 9.16e-1 (4.92e-3) − | 9.59e-1 (1.32e-2) |
| WFG3 | 12 | 2.71e-1 (2.63e-2) − | 3.77e-1 (5.82e-3) − | 2.97e-1 (1.68e-2) − | 3.53e-1 (9.30e-3) − | 2.57e-1 (4.44e-2) − | 3.61e-1 (1.62e-2) − | 4.78e-1 (2.11e-2) |
| WFG4 | 12 | 4.91e-1 (1.41e-2) − | 5.01e-1 (6.63e-3) − | 5.17e-1 (2.21e-3) − | 5.26e-1 (2.51e-3) − | 5.19e-1 (3.28e-3) − | 5.32e-1 (1.82e-3) − | 9.51e-1 (5.11e-3) |
| WFG5 | 12 | 4.77e-1 (8.04e-3) ≈ | 4.81e-1 (2.91e-3) + | 4.98e-1 (3.27e-3) + | 5.06e-1 (4.35e-3) + | 4.94e-1 (5.37e-3) + | 5.05e-1 (3.61e-3) + | 4.43e-1 (2.63e-2) |
| WFG6 | 12 | 4.45e-1 (1.90e-2) − | 4.41e-1 (1.06e-2) − | 4.65e-1 (7.82e-3) − | 4.72e-1 (1.58e-2) − | 4.55e-1 (2.01e-2) − | 4.80e-1 (1.21e-2) − | 6.01e-1 (1.63e-2) |
| WFG7 | 12 | 4.41e-1 (2.54e-2) − | 5.10e-1 (4.93e-3) − | 5.18e-1 (7.89e-3) − | 5.31e-1 (2.05e-3) − | 5.11e-1 (1.06e-2) − | 5.35e-1 (1.96e-3) − | 7.63e-1 (1.54e-2) |
| WFG8 | 12 | 4.11e-1 (1.73e-2) − | 4.23e-1 (3.71e-3) − | 4.31e-1 (1.01e-2) − | 4.40e-1 (2.75e-3) − | 4.30e-1 (7.56e-3) − | 4.44e-1 (6.64e-3) − | 7.33e-1 (2.24e-2) |
| WFG9 | 12 | 3.87e-1 (5.17e-2) − | 4.84e-1 (7.56e-3) − | 4.95e-1 (1.15e-2) − | 5.02e-1 (4.90e-3) − | 4.92e-1 (7.97e-3) − | 5.07e-1 (7.91e-3) − | 9.38e-1 (6.11e-3) |
| +/−/≈ | | 6/10/0 | 7/8/1 | 2/12/2 | 4/11/1 | 4/12/0 | 4/11/1 | |

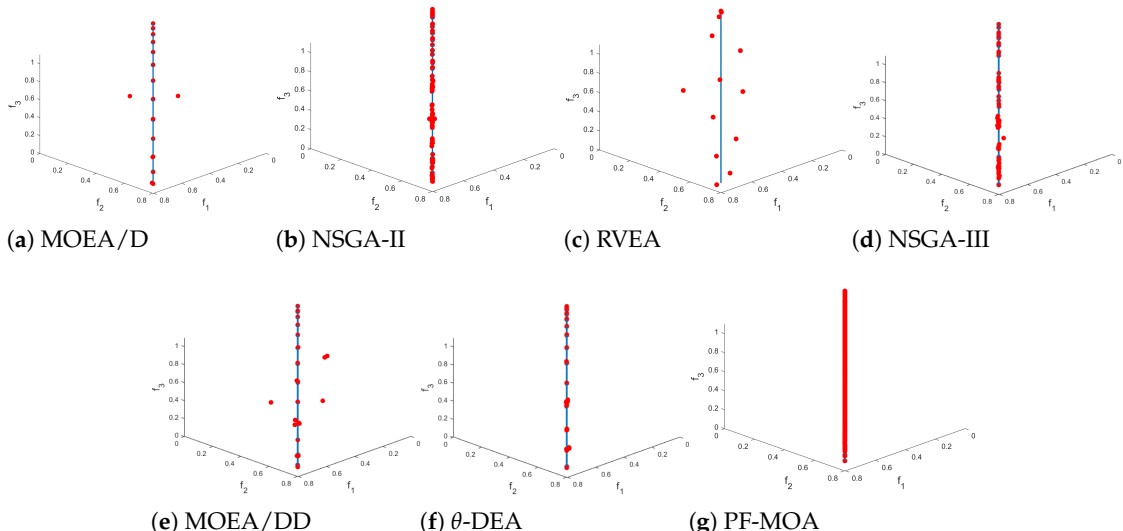

(**a**) MOEA/D  (**b**) NSGA-II  (**c**) RVEA  (**d**) NSGA-III

(**e**) MOEA/DD  (**f**) $\theta$-DEA  (**g**) PF-MOA

**Figure 1.** The Pareto front obtained by the compared algorithms on DTLZ5.

### 5. Conclusions

In this paper, we extended the particle filter optimization method from single-objective optimization to multiobjective optimization. The Tchebycheff decomposition was used to decompose a multi-objective optimization into a set of single-objective problems so that a sequence of target distribution was defined. Subsequently, the particle filter was adopted to simulate these target distributions by using its tracking ability, and genetic operators were employed to perform the particle move. The experimental results on the DTLZ test suite showed the promising performance of PF-MOA compared with three state-of-the-art multi-objective evolutionary algorithms.

However, PF-MOA cannot effectively solve certain problems with discontinuous optimization problems, such as DTLZ6 and DTLZ7. The reason may be that PF-MOA always searches around the best particle, thereby, reducing the diversity of all the particles; however, the lack of diversity cannot be addressed by the resampling step, which should be considered in future work. Moreover, for real-world multiobjective optimization problems, uncertainty is an unavoidable issue, and it directly affects the optimization performance. As the filtering methods have been successfully applied to noisy MOPs, the particle filter may benefit MOEAs for solving MOPs with uncertainty.

**Author Contributions:** Conceptualization, X.W. and Y.J.; methodology, X.W.; software, X.W.; validation, X.W.; formal analysis, X.W.; investigation, X.W.; resources, X.W.; data curation, X.W.; writing—original draft preparation, X.W.; writing—review and editing, X.W. and Y.J.; visualization, X.W.; supervision, Y.J.; project administration, Y.J.; funding acquisition, Y.J. All authors have read and agreed to the published version of the manuscript.

**Funding:** This research received no external funding.

**Conflicts of Interest:** The authors declare no conflict of interest.

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
