# Peer review of "Knowledge Transfer Based on Particle Filters for Multi-Objective Optimization"

_mca, doi:10.3390/mca28010014_

Round 1

Reviewer 1 Report

The topic is interesting, and this paper has its originality.

However, this paper can be improved. Some suggestions are given as follws:

1. The title shows that this paper studies MOA based on transfer learning and particle filter. However, there is no introduction of transfer learning in this paper. It is better to present enough background of transfer learning, and it is necessary to explain the usage of transfer learning fo MOA.

2. It will be better to give the theoretical analysis in the main content of Section 3. However, only algorithms are presented.

3.  The comparative experiments should be fully explained. Only one table and one figure are given without detail explaining. 

4. Some of the cited papers about the particle filter optimzer are more than 15 years ago. In fact, there are some published articles can be found, such as "A multi-objective discrete particle swarm optimization method for particle routing in distributed particle filters".

5. The theoretical introduction of particle filter in Section 2 is necessary, but  the relationship with the main topic of this paper is not introduced. Therefore,  it is difficult for reader to find out the usefullness of these formula to under this paper better.

6. There are some confusing expression in the paper, such as:
  1) In abstract, this sentence is confusing:"As the importance weight updating step takes the previous target distribution as the proposal distribution, and the current target distribution as the target distribution, the knowledge acquired from the previous run can be utilized in the current run by carefully designing the set of target distributions."  What does it mean "the current target distribution as the target distribution"? There are also some sentences in the sections. 
  2) PF is abbreviation of "Pareto Front" in Line 26, while PF is also used as the abbreviation of "Particle Filter" in Line 42. The author should make it identical. 
  3) The paragraph from Line 38 to 65 is too long.
  4) In Algorithm 4, there is words "now particle". It should be "new particle".
  5) The URL of the codes should be given, so that readers could be rerun them and continue the work.

Author Response

Comment 1. The title shows that this paper studies MOA based on transfer learning and particle filter. However, there is no introduction of transfer learning in this paper. It is better to present enough background of transfer learning, and it is necessary to explain the usage of transfer learning fo MOA.

Response: Thanks for your comments. We have explained why knowledge can be transfered from the previous subproblem to the current subproblem in Section 3.3. This is achieved by the nature of particle filter and the set of target distributions. We would like to note that we did not use transfer learning techniques in our work, but the knowledge can be transferred via the way that we use the particle filter. On the other hand, as far as we know, for a standard multi-objective optimization, very few studies have used transfer learning techniques. Instead, transfer learning techniques have been widely used for multi-task optimization, dynamic optimization and so on. Hence, we did not review these transfer learning in the introduction.

Comment 2. It will be better to give the theoretical analysis in the main content of Section 3. However, only algorithms are presented.

Response: Thank you for pointing this. First, it is hard to provide theoretical analysis for evolutionary algorithms. Second, for the particle filter, there exists some theoretical analysis for the design of target distributions and  Metropolis sampling. Hence, we have added the related reference in the revised manuscript. 

3.  The comparative experiments should be fully explained. Only one table and one figure are given without detail explaining. 

Response: Thanks for your helpful comments. More details have been added in the Section 4.4. 

4. Some of the cited papers about the particle filter optimzer are more than 15 years ago. In fact, there are some published articles can be found, such as "A multi-objective discrete particle swarm optimization method for particle routing in distributed particle filters".

Response: Thank you for your helpful comments. Accordingly, I have added the mentioned reference.

5. The theoretical introduction of particle filter in Section 2 is necessary, but  the relationship with the main topic of this paper is not introduced. Therefore,  it is difficult for reader to find out the usefullness of these formula to under this paper better.

Response: Thanks for the comments. Accordingly, we have removed some formulas from Section2. 

6. There are some confusing expression in the paper, such as:
  1) In abstract, this sentence is confusing:"As the importance weight updating step takes the previous target distribution as the proposal distribution, and the current target distribution as the target distribution, the knowledge acquired from the previous run can be utilized in the current run by carefully designing the set of target distributions."  What does it mean "the current target distribution as the target distribution"? There are also some sentences in the sections. 

Response: Thank you for your comments. At first, we design a set of target distributions, each one of which is corresponding to a subproblem, and the particle filter is used to approximate these distributions one by one. Due to the feature of the particle filter, we need to push these particles to move from one distribution to another. So the importance weight updating step takes the previous target distribution as the proposal distribution, and the current target distribution as the target distribution.More details can be found in Section 3.3.

  2) PF is abbreviation of "Pareto Front" in Line 26, while PF is also used as the abbreviation of "Particle Filter" in Line 42. The author should make it identical. 

Response: Thsn you for pointing this out. Accordingly, we have corrected our abbreviations.

  3) The paragraph from Line 38 to 65 is too long.

Response: Thank you for your comments, and the sentence has been revised accordingly.

  4) In Algorithm 4, there is words "now particle". It should be "new particle".

Response: Sorry for the typo and we have corrected this.

  5) The URL of the codes should be given, so that readers could be rerun them and continue the work.

Response: Thank you for the comments. We have provided a link for the code in the revised manuscript.

Reviewer 2 Report

This paper applied particle filters to the MOEA/D framework and proposed an algorithm called PF-MOA for multi-objective optimization. My main comments are as follows:

(1) In the title and abstract of this paper, knowledge transfer is specifically mentioned in algorithm design. However, I did not find further explanations in the manuscript for the usage and functionality of the transfer learning method. If the Bayesian process is considered a knowledge transfer, then the contribution of this paper may not be as good as it has indicated.

(2) In the introduction, the description of the model-based optimization algorithm does not include the most-recognized CMA-ES and some similar evolutionary strategy, which is based on the covariance matrix adaptation.

(3) The comparison algorithms such as NSGA-II is not the latest one, the following NSGA-III with much better performance than NSGA-II should be compared. Moreover, some state-of-the-art algorithms, such as MOEA/DD and theta-EDA are not included. To provide an unbiased comparison, the HV metric is also needed in experimental studies. Moreover, the model-based optimization algorithm is also not included in the comparison.

(4) In the pseudo of algorithm 3, there is an undefined indicator in “weight vectors ˘ for the Tchebycheff approach”.

Author Response

Comment (1) In the title and abstract of this paper, knowledge transfer is specifically mentioned in algorithm design. However, I did not find further explanations in the manuscript for the usage and functionality of the transfer learning method. If the Bayesian process is considered a knowledge transfer, then the contribution of this paper may not be as good as it has indicated.

Response: Thanks for your comments. We have explained why knowledge can be transfered from the previous subproblem to the current subproblem in Section 3.3. This is achieved by the nature of particle filter and the set of target distributions. We would like to note that we did not use transfer learning techniques in our work, but the knowledge can be transferred via the way that we use the particle filter. From the perspective of multi-objective optimization, the advantage of the proposed PF-MOA can be explained by tracking the Pareto optimal solutions on the Pareto front and making the search more efficiently. The reason is that the importance weight of particles in the proposed PF-MOA is updated according to the difference between the current and the previous distributions, which correspond to two related subproblems.

(2) In the introduction, the description of the model-based optimization algorithm does not include the most-recognized CMA-ES and some similar evolutionary strategy, which is based on the covariance matrix adaptation.

Response: Thank you for pointing this out. Accordingly, we have added the CMA-ES in the description of the model-based optimization.

(3) The comparison algorithms such as NSGA-II is not the latest one, the following NSGA-III with much better performance than NSGA-II should be compared. Moreover, some state-of-the-art algorithms, such as MOEA/DD and theta-EDA are not included. To provide an unbiased comparison, the HV metric is also needed in experimental studies. Moreover, the model-based optimization algorithm is also not included in the comparison.

Response: Thank you for your comments. Based on your comments, we have added NSGA-III and MOEA/DD to the comparison. As we did not find the code and original paper for theta-EDA, we guess that the reviewer maybe mean the theta-DEA, so we did not include theta-EDA, inseatd including theta dominance relation-based evolutionary algorithm for many-objective optimization (theta-DEA)  in the comparison.  

(4) In the pseudo of algorithm 3, there is an undefined indicator in “weight vectors ˘ for the Tchebycheff approach”.

Response: Thank you for pointing this out. The weight vector is defined in Eq. 19, and the corresponding description can be found in Section 3.2. While the indicator j denotes the j-th weight vector, the indicator i denote the i-th dimension of the weight vector. 

Reviewer 3 Report

This article incorporate transfer learning capabilities into the optimizer by using particle filters and propose a novel particle filter based multi-objective optimization algorithm (PF-MOA) by transferring knowledge acquired from the search experience.

The ideas are interesting, and the performance is promising. The paper is well written and rich in work. The related research has some creative and reference value. I also have some suggestions:

 1. Some literature relevant to this work of recent years should be cited.

 2. The article states “The way of importance updating makes it possible to leverage the knowledge readily available for the previous subproblem to optimize the next subproblem”. If the differences between each problem in a multi-objective optimization are significant, will using knowledge from the previous subproblem to update another problem lead to poorer results?

3. What are the advantages of using "the Metropolis sampling method associated with genetic operators is adopted to sample new particles".

 4. The analysis of the experimental results is not detailed enough, for example, the results on WFG5, WFG8 are not as good as the other three comparison algorithms, what are the possible reasons.

 5.The analysis of the figure (Fig 1) is not detailed enough.

 6.The structure of the article needs to be adjusted. Some shortcomings of the research related to the article, current problems, and innovations and contributions of your work should be listed.

 7.The parameter settings about the algorithm and the experiment should be given in the paper.

 8.What is the relevance of the three comparison methods selected to the work in this paper and why they were chosen for comparison needs to be explained in the paper.

 9.1-3 contrasting methods should be added to the paper.

Author Response

Comment1: Some literature relevant to this work of recent years should be cited.

Response: Thank you for your helpful comments. Accordingly, we have added the related work in the Introduction.

Comment2: The article states “The way of importance updating makes it possible to leverage the knowledge readily available for the previous subproblem to optimize the next subproblem”. If the differences between each problem in a multi-objective optimization are significant, will using knowledge from the previous subproblem to update another problem lead to poorer results?

Response: Thank you for pointing this out. According to our experimental results, the answer to this question is yes. For example, for DTLZ6 and DTLZ7, the performance of the proposed algorithm is poor. To alleviate this problem, the search ability of the proposed algorithm should be further improved, which can be considered in our future work.

Comment 3: What are the advantages of using "the Metropolis sampling method associated with genetic operators is adopted to sample new particles".

Response: Thank you for the constructive comments. The resampling step together with the genetic operator based Metropolis sampling step prevents sample degeneracy, or in other words, keeps the sample diversity and thus the exploration of the solution space. Genetic operators are used here because they are widely used and effective.

Comment 4: The analysis of the experimental results is not detailed enough, for example, the results on WFG5, WFG8 are not as good as the other three comparison algorithms, what are the possible reasons.

Response: Thank you for your comments. Accordingly, we have added more details in terms of the analysis of the experimental results.

 Comment 5.The analysis of the figure (Fig 1) is not detailed enough.

Response: Thank you for your comments. More analysis have been added in the manuscript for Fig. 5. 

 Comment 6.The structure of the article needs to be adjusted. Some shortcomings of the research related to the article, current problems, and innovations and contributions of your work should be listed.

Response: Thank you for your comments. Accordingly, we have re-organized the Introduction. 

 Comment 7.The parameter settings about the algorithm and the experiment should be given in the paper.

Response: We would like to note that the parameter settings are given in Section 4.3.

 Comment 8. What is the relevance of the three comparison methods selected to the work in this paper and why they were chosen for comparison needs to be explained in the paper.

Response: Thank you for pointing this out. Accordingly, we have added the reason for the selection of these compared algorithms. Moreover, we have included more algorithms that are state-of-the-art and representative multi-objective evolutionary algorithms in the comparison.

 Comment 9.1-3 contrasting methods should be added to the paper.

Response: Thanks for your comments. We have included more algorithms in the comparison based on the reviewer's comments.

Reviewer 4 Report

The paper presented an algorithm incorporating transfer learning into particle filtering optimization(PFO) method in order to extend its capabilities in the multi-objective settings. The algorithm, together with its motivation and some theoretical backgrounds, are well presented. The experiments by comparing the proposed algorithm with some canonical methods used in multi-objective optimizations exemplifies that the proposed algorithm is at least comparable to those canonical methods. 

Overall, the algorithm proposed in this paper is novel and very well formulated, and the numerical study is relatively sufficient. However, some theoretical arguments of the algorithm are lack in the paper, which weaken the significance of the contribution of the paper. Some theoretical arguments are, for example, the convergence of the algorithm to the optimal and the complexity analysis of the algorithm. Therefore, the reviewer is expecting more theoretical discussions of the paper regarding the above to aspects before approve an acceptance. 

Author Response

Comment1: The paper presented an algorithm incorporating transfer learning into particle filtering optimization(PFO) method in order to extend its capabilities in the multi-objective settings. The algorithm, together with its motivation and some theoretical backgrounds, are well presented. The experiments by comparing the proposed algorithm with some canonical methods used in multi-objective optimizations exemplifies that the proposed algorithm is at least comparable to those canonical methods. 

Response: Thank you for your encouraging comments.

Comment 2: Overall, the algorithm proposed in this paper is novel and very well formulated, and the numerical study is relatively sufficient. However, some theoretical arguments of the algorithm are lack in the paper, which weaken the significance of the contribution of the paper. Some theoretical arguments are, for example, the convergence of the algorithm to the optimal and the complexity analysis of the algorithm. Therefore, the reviewer is expecting more theoretical discussions of the paper regarding the above to aspects before approve an acceptance.

Response: Thank you for the helpful comments. It is hard to provide theoretical analysis for evolutionary algorithms, but we have tried to provide more theoretical discussions for the particle filter part in the revised manuscript.

Round 2

Reviewer 3 Report

The authors have revised the manuscript and the current version can be accepted with no further revision.

Reviewer 4 Report

The reviewer doesn't have any concerns for the revised version, will accept the current form.